



# Comparing different generations of idealized solar geoengineering simulations in the Geoengineering Model Intercomparison Project (GeoMIP)

Ben Kravitz[1,2], Douglas G. MacMartin[3], Daniele Visioni[3], Olivier Boucher[4], Jason N. S. Cole[5], Jim Haywood[6,7], Andy Jones[7], Thibaut Lurton[4], Pierre Nabat[8], Ulrike Niemeier[9], Alan Robock[10], Roland Séférian[8], and Simone Tilmes[11]

[1]Department of Earth and Atmospheric Sciences, Indiana University, Bloomington, IN, USA
[2]Atmospheric Sciences and Global Change Division, Pacific Northwest National Laboratory, Richland, WA, USA
[3]Sibley School of Mechanical and Aerospace Engineering, Cornell University, Ithaca, NY, USA
[4]Institut Pierre-Simon Laplace (IPSL), Sorbonne Université/CNRS, Paris, France
[5]Environment and Climate Change Canada, Toronto, Ontario, Canada
[6]College of Engineering, Mathematics and Physical Sciences, University of Exeter, Exeter, United Kingdom
[7]UK Met Office Hadley Centre, Exeter, United Kingdom
[8]CNRM, Université de Toulouse, Météo-France, CNRS, Météo-France, Toulouse, France
[9]Max Planck Institute for Meteorology, Hamburg, Germany
[10]Department of Environmental Sciences, Rutgers University, New Brunswick, NJ, USA
[11]Atmospheric Chemistry Observations and Modeling Laboratory, National Center for Atmospheric Research, Boulder, CO, USA

**Correspondence:** Ben Kravitz, 1001 E. 10th Street, Bloomington, IN 47405-1405, USA. (bkravitz@iu.edu)

**Abstract.** Solar geoengineering has been receiving increased attention in recent years as a potential temporary solution to offset global warming. One method of approximating global-scale solar geoengineering in climate models is via solar reduction experiments. Two generations of models in the Geoengineering Model Intercomparison Project (GeoMIP) have now simulated offsetting a quadrupling of the $CO_2$ concentration with solar reduction. This simulation is artificial and designed to elicit large

responses in the models. Here we show that energetics, temperature, and hydrological cycle changes in this experiment are statistically indistinguishable between the two ensembles. Of the variables analyzed here, the only major differences involve highly parameterized and uncertain processes, such as cloud forcing or terrestrial net primary productivity. We conclude that despite numerous structural differences and uncertainties in models over the past 20 years, including an increase in climate sensitivity in the latest generation of models, broad conclusions about the climate response to global solar dimming remain

robust.



# 1   Introduction

Solar geoengineering describes a set of technologies designed to (ideally) temporarily, deliberately reduce some of the effects of climate change by changing the radiative balance of the planet, often by reflecting sunlight back to space (NRC, 2015).

Numerous methods have been proposed, but the most studied is stratospheric sulfate aerosol injection (Budyko, 1977; Crutzen, 2006). This method involves substantially increasing the stratospheric sulfate aerosol burden, replicating the mechanisms that cause cooling after large volcanic eruptions (Robock, 2000). Climate models remain the most promising tools for understanding the consequences of solar geoengineering. In model simulations of solar geoengineering, insolation reduction is often used as a proxy for actual stratospheric sulfate aerosols, as it captures many of the broad radiative effects of stratospheric aerosol

geoengineering as well as some of the important climate effects like surface cooling and hydrological cycle strength reduction (Niemeier et al., 2013; Kalidindi et al., 2015). However, stratospheric sulfate aerosols also absorb longwave radiative flux, which heats the upper troposphere and lower stratosphere. As such, any implementation of stratospheric geoengineering with sulfate aerosols would produce additional effects, such as changing atmospheric circulation in response to stratospheric heating and heating gradients (e.g., Richter et al., 2017; Tilmes et al., 2018; Simpson et al., 2019) and stratospheric ozone changes (e.g.,

Pitari et al., 2014), as well as changes in ultraviolet radiative flux and enhanced diffuse radiation at the surface (Madronich et al., 2018). However, here we consider the major, large-scale effect of reflecting sunlight to cool Earth.

Simulations of solar geoengineering with solar reduction have long shown that solar geoengineering would cool the planet, offsetting global warming (e.g., Govindasamy and Caldeira, 2000; NRC, 2015; Irvine et al., 2016). Idealized simulations of solar reduction have also been simulated in a multi-model context under the Geoengineering Model Intercomparison Project

(GeoMIP; Kravitz et al., 2011), to understand the robust model responses to various standardized solar geoengineering simulation designs. Multi-model conclusions from these studies indicate that solar geoengineering would be effective at partially offsetting greenhouse gas-induced temperature changes (Kravitz et al., 2013), as well as changes in the hydrological cycle (Tilmes et al., 2013), the cryosphere (Moore et al., 2014), extreme events (Curry et al., 2014; Aswathy et al., 2015), vegetation (Glienke et al., 2015), circulation (Guo et al., 2018; Gertler et al., 2020), agriculture (Xia et al., 2014), and numerous

other areas. However, the offset is not perfect (Moreno-Cruz et al., 2012), particularly on a regional basis or when considering multiple simultaneous metrics of climate change (Kravitz et al., 2014; Irvine et al., 2019), leading to concerns about winners and losers from geoengineering (Ricke et al., 2010). To some extent, the effects of solar geoengineering may be tailored or designed (MacMartin et al., 2013; Kravitz et al., 2016, 2017, 2019), but solar geoengineering will still not be able to perfectly offset climate change from greenhouse gases.

The previous phase of GeoMIP was associated with the Coupled Model Intercomparison Project Phase 5 (CMIP5; Taylor et al., 2012), an international collaboration of climate models to attempt to understand robust model responses to various forcings. GeoMIP has now entered a new phase, concurrent with the Coupled Model Intercomparison Project Phase 6 (CMIP6; Eyring et al., 2016), and with it are new solar geoengineering simulations with new and updated versions of Earth System Models (Kravitz et al., 2015). As such, this is an opportunity to revisit some central questions in solar geoengineering. Many

of the CMIP5 results regarding solar geoengineering showed substantial agreement across the participating GeoMIP models.



In this newest iteration of GeoMIP, do the same science conclusions still hold, and do the models still generally agree on the resulting climate effects? Here we address these questions by evaluating and comparing general climate model response to GeoMIP experiment G1 (described in the next section) from both CMIP5 and CMIP6.

## 2  Simulations and Participating Models

In this study, we evaluate GeoMIP experiment G1, in which, starting from a preindustrial control (piControl) baseline, the atmospheric $CO_2$ concentration is instantaneously quadrupled (the standard CMIP experiment abrupt4xCO2), and insolation is simultaneously reduced such that net top-of-atmosphere (TOA) radiative flux is approximately unchanged from the baseline in the first decade of simulation (Kravitz et al., 2011, 2015). This experiment was part of the original suite of GeoMIP experiments and was repeated and extended in the newest suite in an effort to understand the role of model structural uncertainty in broad

conclusions about solar geoengineering. Participating models are listed in Table 1. We include 13 models from CMIP5 and 7 models from CMIP6.

The original G1 experiment was 50 years in length, whereas the CMIP6 version is 100 years in length to allow for better analyses of rare events or to capture very slow responses. Comparison between the two ensembles necessitates only using the first 50 years, but we need to verify that this can be done without losing important longer-term evolution in features. Figures 1

and 2 look at G1 behavior over the entire 100-year period of the CMIP6 simulations to determine whether there is any drift or trend that would not be revealed by only analyzing 50 years. With the exception of IPSL-CM6A-LR, no model shows any long-term behavior in temperature. Two models (IPSL-CM6A-LR and GISS-E2.1-G) show a slight trend in precipitation and evaporation, with a change of <1% over the first 50 years of simulation. As such, we conclude that our choice to focus on the first 50 years of simulation does not appreciably affect our results.

Figure 2 indicates that the temperature trend in IPSL-CM6A-LR is due to temperature changes north of 30°N, possibly related to a slight trend in sea ice coverage (Boucher et al., 2020). This model is also known to have a bicentennial oscillation, which could affect G1–piControl differences, depending on the baseline period used for subtraction. To verify that this oscillation is not impacting our results, we divided that model's 1200-year piControl run into 50-year chunks and computed the surface air temperature average for each of those chunks. The largest temperature found was 286.0339 K, and the smallest

was 285.6384 K. The average over the entire ensemble was 285.8604 K. As such, using the mean of the entire ensemble versus matching the appropriate period in the bicentennial oscillation would have an impact on G1–piControl temperature by at most 0.22 K. Only averaging the first 100 years of the piControl run (which may be the best match to the period covered by G1) yields a temperature of 285.9084 K, which is 0.048 K different from the mean of the entire piControl run. As such, we conclude that this bicentennial oscillation is unlikely to have substantially influenced our findings.

Because the main focus of this paper is a comparison between the CMIP5 and CMIP6 generations of model results, we have opted for the following to aid comparisons:





- Since we are not evaluating any features that require 100 years of statistics, and the results do not show any appreciable time evolution of behavior after the first couple of years (see discussion above), we only evaluate the first 50 years of all simulations. All maps show changes over years 11-50, removing the initial transient period.

- We do not compare previous versions of individual models with current ones, instead only examining ensembles. Even though models may share similar development histories (e.g., atmosphere and ocean dynamical cores, convective parameterizations, radiative transfer modules, terrestrial biosphere and cryosphere; Knutti et al., 2013; Zelinka et al., 2020), there have been numerous developments in models in these areas (and others) between CMIP5 and CMIP6 such that in most cases a direct comparison would not be meaningful.

- We focus extensively on the G1 results and, with few exceptions, do not focus on the corresponding abrupt4xCO2 simulations. It has been well documented that the CMIP6 models tend to have higher climate sensitivities than the CMIP5 models (Flynn and Mauritsen, 2020; Meehl et al., 2020; Zelinka et al., 2020), so we do not wish to make conclusions that might be based on a form of selection bias.

- All lack of stippling on map plots, as in previous GeoMIP studies (e.g., Kravitz et al., 2013), indicates agreement on the
sign of the response in at least 75% of models. Because $G1_{CMIP5}$ has more participating models than $G1_{CMIP6}$, this threshold provides some consistency across analyses of the ensembles. When plotting differences between the ensembles ($G1_{CMIP6}$–$G1_{CMIP5}$), there is no stippling, as it is difficult to meaningfully represent such differences between ranges. Aggregate differences between the two ensembles, as calculated using Welch's $t$-test or differences in stippled area, are discussed in Table 2.

## 3   Results

### 3.1   Energetics

Ensemble mean radiative and turbulent flux quantities are plotted in Figure 3, and the ensemble ranges are plotted in Figure 4. An immediate observation is that, in both ensembles, the models were successful at limiting net TOA radiative flux change to within approximately $\pm 0.1 \, \mathrm{W \, m^{-2}}$ of the models' respective preindustrial values. Accomplishing this required an average solar
reduction of 4.14% (models range in 3.20–5.00%) in CMIP5 and 4.14% (3.72–4.91%) in CMIP6. As such, despite numerous structural changes between the two generations of models, there is no appreciable change in solar efficacy (Hansen et al., 2005).

None of the radiative flux quantities indicate large transients over 50 years of simulation of G1, other than the initial flux change within the first year or so of simulation. This is consistent with the "perpetual fast response" found by Kravitz et al. (2013b), in which because global mean temperature does not change appreciably over the course of the G1 simulation, climate
feedbacks are not excited, and the internal state of the system (as measured by, for example, fluxes and hydrological cycle changes) similarly does not change. Ensemble mean fluxes show few differences ($<1 \, \mathrm{W \, m^{-2}}$ in magnitude) with the exception of shortwave cloud forcing, defined as all-sky minus clear-sky shortwave flux at the surface. On average, the CMIP6 ensemble



has 3–4 W m$^{-2}$ less shortwave cloud forcing than CMIP5. Neglecting some outliers, for each flux except shortwave (and hence total) cloud forcing, the median model in one ensemble is within the inter-quartile range of the other ensemble. This

indicates that there are no major differences between the ensembles in how the models handle energy balance and energetics, with the exception of clouds, which is consistent with findings about CMIP6 (Zelinka et al., 2020). Moreover, it appears that most of the major differences in shortwave cloud forcing are due to outliers in each ensemble, positive for CMIP5 and negative for CMIP6. To further explore these potential differences, Figure 5 provides maps of the ensemble means for cloud forcing. In G1, the CMIP5 ensemble showed more positive shortwave cloud forcing and more negative longwave cloud forcing (i.e.,

more cancellation) than the CMIP6 ensemble. Overall, the CMIP6 ensemble has greatly reduced (in some places by over 10 W m$^{-2}$) shortwave cloud forcing as compared to CMIP5 under the G1 experiment. This is a widespread result, but the most prominent features are in the tropics, especially over the Amazon, Africa, and the Maritime Continent. These regions encompass tropical forests, indicating a potential for vegetation feedbacks on the temperature reductions. However, the reasons behind these forcing changes are difficult to diagnose, as they could be due to changes in cloud thickness, cloud cover, or cloud

level between CMIP5 and CMIP6 models (e.g., Vignesh et al., 2020), differences in how solar geoengineering affects clouds (Russotto and Ackerman, 2018), or artifacts of the analyses (e.g., cloud masking; Andrews et al., 2009; Kravitz et al., 2013b). Moreover, based on the results in Figure 4, it is likely that many of these features are exaggerated by outlier models (also see Vignesh et al., 2020). As such, we reserve such detailed investigations for future work.

### 3.2 Temperature

These small flux changes also lead to few G1 temperature changes between the two ensembles. Figure 6 shows global, land, and ocean-averaged temperatures for the CMIP5 and CMIP6 ensembles. In general, the abrupt4xCO2 simulation in CMIP6 has higher temperatures than in CMIP5, consistent with the noted increase in climate sensitivity (Vial et al., 2013; Flynn and Mauritsen, 2020; Meehl et al., 2020; Zelinka et al., 2020). In both ensembles, G1 is effective at offsetting global mean temperature change, in some cases with a slight positive residual temperature change over land. Figure 7 shows three aggregate

temperature metrics: global mean temperature ($T_0$), the interhemispheric temperature gradient ($T_1$), and the equator-to-pole temperature gradient ($T_2$) (Ban-Weiss and Caldeira, 2010; Kravitz et al., 2016):

$$T_0 = \frac{1}{A} \int_{-\pi/2}^{\pi/2} T(\psi)\,\mathrm{d}A$$

$$T_1 = \frac{1}{A} \int_{-\pi/2}^{\pi/2} T(\psi)\sin\psi\,\mathrm{d}A \tag{1}$$

$$T_2 = \frac{1}{A} \int_{-\pi/2}^{\pi/2} T(\psi)\frac{1}{2}(3\sin^2\psi - 1)\,\mathrm{d}A$$



where $A$ is area. As for the fluxes, the median model in one ensemble is within the inter-quartile range of the other ensemble. This indicates that no ensemble is on average warmer or cooler than another, has a substantially warmer Northern or Southern

Hemisphere than the other, or has warmer tropics or poles than the other. We can conclude that spatial patterns of temperature change from G1 are robust across a wide range of structural uncertainty, including an increase in climate sensitivity between the two generations of CMIP.

The spatial structure of temperature change (Figure 8) does have small differences between the two ensembles. G1 in CMIP6 has multiple locations that are warmer than G1 in CMIP5, despite both ensembles achieving net energy balance at TOA and

the surface (Figure 3). The majority of the differences are over land and in the tropics, where CMIP6 is slightly warmer than CMIP5 (up to $1°C$ in some places). Nevertheless, both ensembles show the well noted feature that offsetting a $CO_2$ increase with globally uniform solar reduction overcools the tropics and undercools the poles (Govindasamy and Caldeira, 2000; Kravitz et al., 2013). CMIP6 shows slightly less high latitude warming than CMIP5, but temperature differences between the two ensembles are largely negligible. However, the warmer temperatures in CMIP6 near Greenland have important implications

for ice sheet melt and consequent sea level rise, as well as bottom water formation. We reserve such analyses for future investigations, particularly since the models used here are not capable of simulating the eustatic component of sea level rise. In any case, these ensemble mean differences between CMIP5 and CMIP6 cannot be deemed statistically significant (Table 2 and Figure 7).

### 3.3 Hydrological and Other Integrative Changes

Figure 9 shows ensemble mean changes in precipitation (P), evaporation (E), and P–E for $G1_{CMIP5}$ and $G1_{CMIP6}$. Qualitatively, patterns are similar between both ensembles. Precipitation is slightly (<0.3 mm/day in magnitude) different in the tropics between the two ensembles. The majority of those features can be summarized as a more southward Intertropical Convergence Zone (ITCZ), more precipitation in the South Pacific Convergence Zone, and less precipitation over Southeast Asia and the Maritime Continent in $G1_{CMIP6}$. Evaporation in the two ensembles is nearly identical except for more evaporation in Amazo-

nia and Australia in $G1_{CMIP6}$. As such, the net P–E change between the two ensembles strongly resembles the precipitation changes. Figure 10 shows that, like previous evaluations of ensemble ranges, the median model in one ensemble falls well within the interquartile range of the other ensemble for P, E, and P–E. As such, we cannot conclude any robust hydrological cycle changes between the two ensembles.

Figure 11 shows average (years 11–50) temperature change (with respect to piControl) plotted against average precipitation

change for each model, as in Tilmes et al. (2013). Other than a potentially greater climate sensitivity of some CMIP6 models, there is no distinguishable difference in aggregate behavior between the two ensembles. The same conclusion discovered by Tilmes et al. (2013) holds: solar reduction cannot simultaneously offset $CO_2$-induced changes in both global mean temperature and global mean precipitation.

As an integrator of $CO_2$, temperature, and precipitation effects over land, Figure 12 shows changes in terrestrial net primary

productivity (NPP). Numerous land regions have lower NPP in CMIP6 than in CMIP5. The ensemble average global NPP change (G1–piControl) is 51.2 (4.1–122.1) Pg C $y^{-1}$ in CMIP5 and 38.1 (19.5–77.5) Pg C $y^{-1}$ in CMIP6, representing a 25.6%





difference in means. Jones et al. (2013) used NPP to highlight the importance of understanding the influence of structural land model differences on climate results related to geoengineering. While it is beyond the scope of this study to perform a detailed diagnosis of which uncertainties or processes are responsible for this inter-ensemble difference, we show that the ensemble

spread of total terrestrial NPP is smaller in CMIP6 than in CMIP5. This result is consistent with the recent assessment of carbon cycle feedbacks conducted by Arora et al. (2020), which demonstrates that the CMIP6 ensemble has reduced overall uncertainty in the land carbon cycle to rising $CO_2$ compared to their CMIP5 predecessors.

## 4   Discussion and Conclusions

Based on the results presented here, model response to G1 has not changed substantially between CMIP5 and CMIP6, despite

numerous changes to models between the two generations, including an increase in climate sensitivity. The sign of residual climate impacts (for example in temperature) are in better agreement in CMIP5 than CMIP6 (Table 2 shows a difference in stippled area between the two ensembles), but this could be a function of the smaller ensemble size in CMIP6. Energetics, temperature, and the hydrological cycle are qualitatively and quantitatively similar in both ensemble means and ensemble ranges, although these variables are somewhat related, so we might expect them to all portray a similar picture. Notable

differences do exist in shortwave cloud forcing and NPP, particularly in Amazonia, Africa, and Australia, which are also regions of inter-ensemble difference in precipitation.

From these findings, we can conclude that results obtained over the past 20 years of study have not been overturned by the latest round of simulations. All of the major ensemble differences highlighted above deal with clouds and land surface modeling, both of which are difficult to model and are necessarily highly parameterized. The conclusions that are based on more

fundamental knowledge, such as column energetics (in the case of the hydrological cycle), are relatively robust to structural uncertainty, in so far as this study adequately captures representative variations in structural uncertainty. This lends confidence to our conclusions, especially regarding robustness to uncertainty, about the broad climate effects from solar geoengineering methods that can be accurately represented via solar dimming.

We also conclude that the models used in CMIP5 are not obviously biased or inferior as compared to CMIP6. While im-

provements have been made in the CMIP6 generation of models, and those models are likely better for representing numerous features of the present-day climate that may be important for studies of geoengineering, there are many aspects of climate that are well represented by earlier models. In some cases, more robust analyses may be enabled by augmenting ensemble sizes with archived output from earlier generations of CMIP models.

Many of the broad features of solar geoengineering with sulfate aerosols can be represented by a reduction in solar constant

(e.g., Niemeier et al., 2013; Kalidindi et al., 2015). However, the more subtle changes that derive from complex response to stratospheric aerosol heating (for example, consequences of stratospheric heating like the positive wintertime North Atlantic Oscillation; Simpson et al., 2019) require detailed assessments with state-of-the-art aerosol microphysical schemes. This is particularly important for understanding regional and seasonal solar geoengineering (Kravitz et al., 2017; Visioni et al., 2019). Such detailed microphysical calculations can only be simulated in a small number of modelsWhile simple G1-style experiments



enable a robust multi-model ensemble analysis, they cannot capture details that depend on microphysics. We emphasize the importance of a variety of modeling approaches to understand solar geoengineering, particularly the role of model uncertainty in conclusions about solar geoengineering.

There are numerous aspects of physical climate that we did not evaluate, nor did we pursue analyses beyond physical climate, including many other aspects of natural science, social science, the humanities, governance, justice, or ethics (to name a few
important areas). Moreover, we emphasize that experiment G1 is an idealized experiment aimed at understanding physical climate response to combinations of large forcings and should not be interpreted as a realistic or policy-relevant scenario of geoengineering. A holistic assessment of the consequences of geoengineering, particularly of more policy-relevant scenarios, would certainly need to take these numerous aspects into account. Nevertheless, based on the results presented here, results for geoengineering across several important metrics appear to be robust to some amount of structural uncertainty. This lends
confidence to some conclusions drawn from global climate models regarding solar geoengineering.

*Data availability.*  All CMIP5 and CMIP6 output, including the respective GeoMIP simulations, is available via the Earth System Grid Federation (https://esgf-node.llnl.gov/projects/esgf-llnl/) or by contacting the respective modeling groups responsible for the output. For CMIP6 output, seeTable 1 data citations.

*Author contributions.*  BK, OB, JNSC, JH, AJ, TL, PN, UN, RS, and ST contributed model output. BK performed the analysis. BK, DGM,
and DV wrote the manuscript with all coauthors.

*Competing interests.*  None.

*Acknowledgements.*  We acknowledge the World Climate Research Programme, which, through its Working Group on Coupled Modelling, coordinated and promoted CMIP. We thank the climate modeling groups for producing and making available their model output, the Earth System Grid Federation (ESGF) for archiving the data and providing access, and the multiple funding agencies who support CMIP6 and
ESGF. We also thank all participants of the Geoengineering Model Intercomparison Project and their model development teams. Support for B.K. was provided in part by the National Science Foundation (NSF) through agreement CBET-1931641, the Indiana University Environmental Resilience Institute, and the Prepared for Environmental Change Grand Challenge initiative. The Pacific Northwest National Laboratory is operated for the US Department of Energy by Battelle Memorial Institute under contract DE-AC05-76RL01830. Resources supporting this work were provided by the NASA High-End Computing (HEC) Program through the NASA Center for Climate Simulation
(NCCS) at Goddard Space Flight Center. A.R. is supported by NSF grants AGS-1617844 and AGS-2017113. U.N. is supported by the German DFG-funded Research Unit VollImpact FOR2820 sub project TI344/2-1 and MPIESM simulation have been performed on the computer of Deutsches Klimarechenzentrum (DKRZ). O.B. and T.L. were supported by the IPSL Climate Graduate School EUR (ANR grant ANR-



11-IDEX-0004 - 17-EURE-0006). The CMIP6 project at IPSL used the HPC resources of TGCC under the allocations 2016-A0030107732, 2017-R0040110492 and 2018-R0040110492 (project gencmip6) provided by GENCI (Grand Équipement National de Calcul Intensif). R.S. and P.N. were supported by the H2020 CONSTRAIN under the grant agreement No 820829 and the Météo-France/DSI supercomputing center.



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



**Table 1.** All participating models in both the CMIP5 and CMIP6 eras of GeoMIP, including references. For G1 solar reduction, the percentage is calculated as the percent change in incident solar irradiance at the top-of-atmosphere between G1 and its respective piControl run. Numbers in the first column correspond to the model numbers in Figure 11.

| # | Model | Generation | Reference | G1 Solar Reduction (%) | Data not available | Data Citations (CMIP6 only) |
|---|-------|------------|-----------|------------------------|--------------------|------------------------------|
| 1 | BNU-ESM | CMIP5 | Ji et al. (2014) | 3.80 | Cloud forcing | |
| 2 | CanESM2 | CMIP5 | Arora et al. (2011) | 4.00 | | |
| 3 | CCSM4 | CMIP5 | Gent et al. (2011) | 4.25 | NPP | |
| 4 | CESM-CAM5.1-FV | CMIP5 | Neale et al. (2010); Hurrell et al. (2013) | 4.70 | | |
| 5 | CSIRO-Mk3L-1.2 | CMIP5 | Phipps et al. (2011, 2012) | 3.20 | Cloud forcing, NPP | |
| 6 | EC-EARTH | CMIP5 | Hazeleger et al. (2011) | 4.12 | Cloud forcing, NPP | |
| 7 | GISS-E2-R | CMIP5 | Schmidt et al. (2014) | 4.47 | | |
| 8 | HadCM3 | CMIP5 | Gordon et al. (2000) | 4.16 | Cloud forcing, NPP | |
| 9 | HadGEM2-ES | CMIP5 | Collins et al. (2011) | 3.88 | | |
| 10 | IPSL-CM5A-LR | CMIP5 | Dufresne et al. (2013); Hourdin et al. (2012) | 3.50 | NPP | |
| 11 | MIROC-ESM | CMIP5 | Watanabe et al. (2008, 2011) | 5.00 | | |
| 12 | MPI-ESM-LR | CMIP5 | Giorgetta et al. (2013); Stevens et al. (2013) | 4.68 | | |
| 13 | NorESM1-M | CMIP5 | Alterskjær et al. (2012); Kirkevåg et al. (2013) | 4.03 | | |
| 14 | CanESM5 | CMIP6 | Swart et al. (2019c) | 3.72 | | Swart et al. (2019a, b); Cole et al. ( |
| 15 | CESM2-WACCM | CMIP6 | Gettelman et al. (2019) | 4.91 | | Danabasoglu (2019c, b, a) |
| 16 | CNRM-ESM2.1 | CMIP6 | Séférian et al. (2019) | 3.72 | | Séférian (2018b, a, c) |
| 17 | GISS-E2.1-G | CMIP6 | Kelley et al. (2020) | 4.13 | | NASA Goddard Institute for Space |
| 18 | IPSL-CM6A-LR | CMIP6 | Boucher et al. (2020); Lurton et al. (2020) | 4.10 | | Boucher et al. (2018c, b, a) |
| 19 | MPI-ESM1.2-LR | CMIP6 | Mauritsen et al. (2019) | 4.57 | | Wieners et al. (2019a, b) |
| 20 | UKESM1.0-LL | CMIP6 | Sellar et al. (2019) | 3.80 | | Tang et al. (2019b, a); Jones (2019) |





**Table 2.** Ensemble differences between the CMIP5 and CMIP6 ensembles for each variable evaluated in this study (left column). Column 2 indicates the difference between the ensembles in how much of the Earth's surface is not stippled (more than 75% of models agree on the sign of the response; positive values indicate that CMIP6 has more unstippled area than CMIP5). Column 3 indicates the fraction of the Earth's surface for which the CMIP5 ensemble is statistically different from the CMIP6 ensemble, based on 95th percentile confidence intervals from Welch's $t$-test.

| Variable | Stippling (%) | Welch's $t$-test (%) | Notes |
|---|---|---|---|
| Surface air temperature | -25.77 | 0.87 | |
| Precipitation | -3.56 | 11.17 | |
| Evaporation | -2.33 | 6.47 | |
| P–E | -15.23 | 1.13 | |
| SW Cloud Forcing | -8.02 | 9.65 | |
| LW Cloud Forcing | 11.99 | 6.57 | |
| Net Primary Productivity | -1.42 | 1.15 | Land surface only |



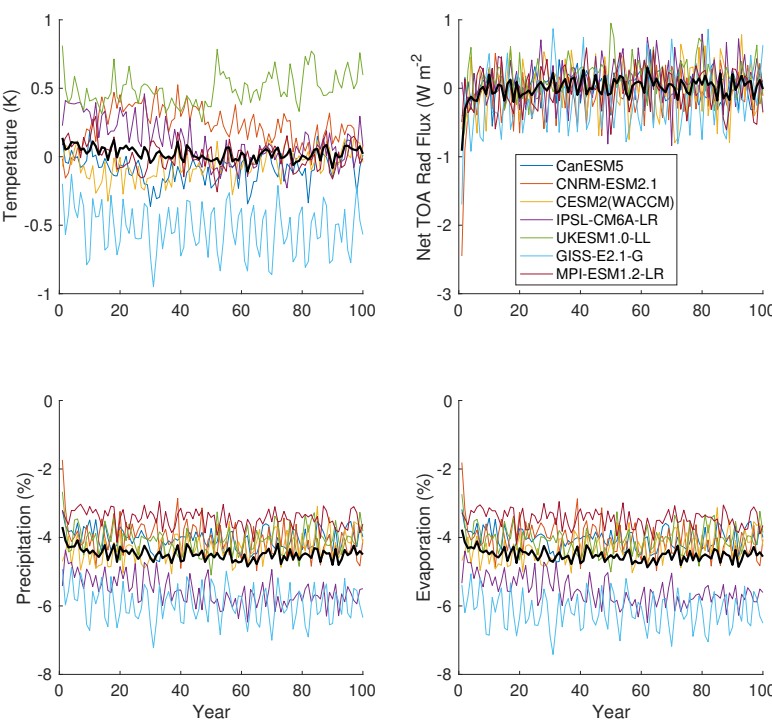

**Figure 1.** Temperature (top left; K), net top-of-atmosphere radiative flux (top right; W m$^{-2}$), precipitation (bottom left; %), and evaporation (bottom right; %) change in G1$_{\mathrm{CMIP6}}$ compared to piControl over 100 years of simulation. Thin colored lines are individual models, and thick black lines are ensemble means.

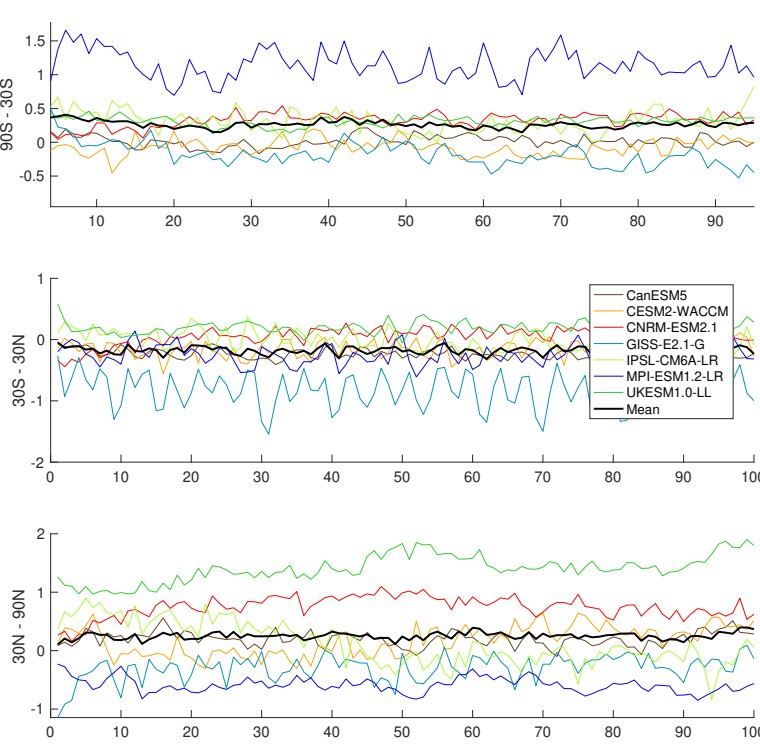

**Figure 2.** Annual mean surface temperature (K) in each model averaged over 90°S-30°S (top), 30°S-30°N (middle), and 30°N-90°N (bottom). The ensemble mean is plotted as thick black lines.

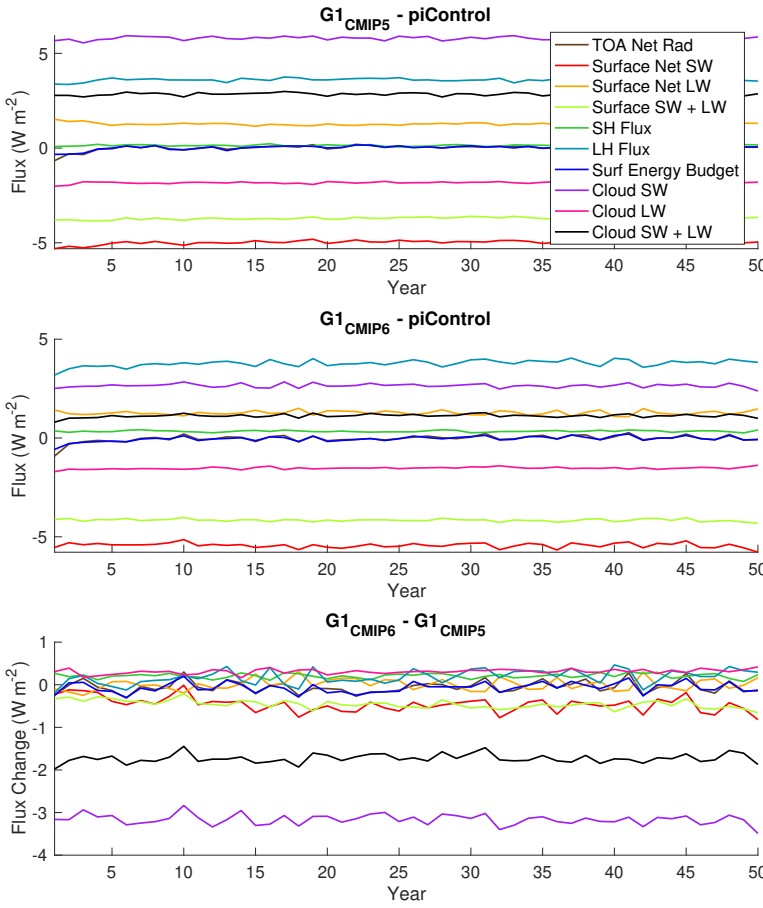

**Figure 3.** Ensemble mean energetics (W m$^{-2}$) for various flux quantities in G1$_{\mathrm{CMIP5}}$ (top), G1$_{\mathrm{CMIP6}}$ (middle), and the difference (bottom). All fluxes are positive downward, which is counterintuitive for sensible heat (SH) and latent heat (LH). Surf Energy Budget indicates the sum of surface shortwave (SW), surface longwave (LW), SH, and LH. Cloud forcing is calculated as all-sky minus clear-sky.



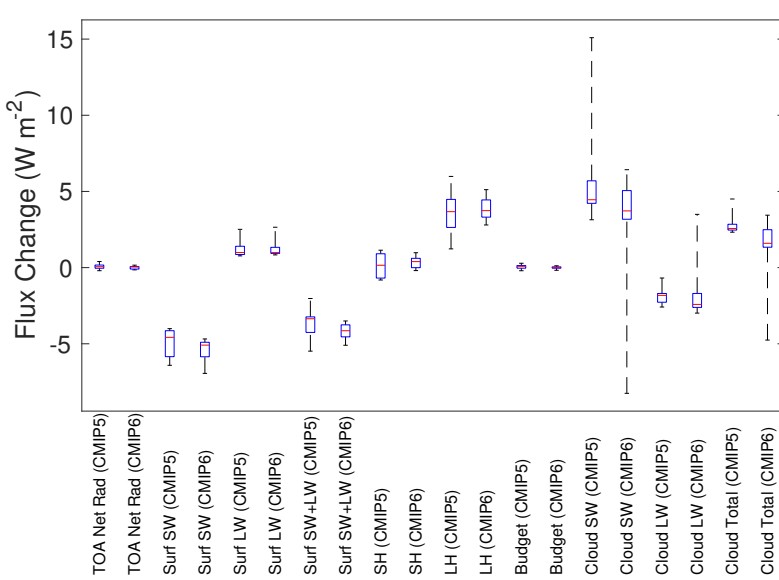

**Figure 4.** Ensemble median (red lines), inter-quartile (blue boxes), and ranges (black whiskers) for the same global mean energetics quantities as in Figure 3 (G1 minus piControl) for both the CMIP5 and CMIP6 ensembles.



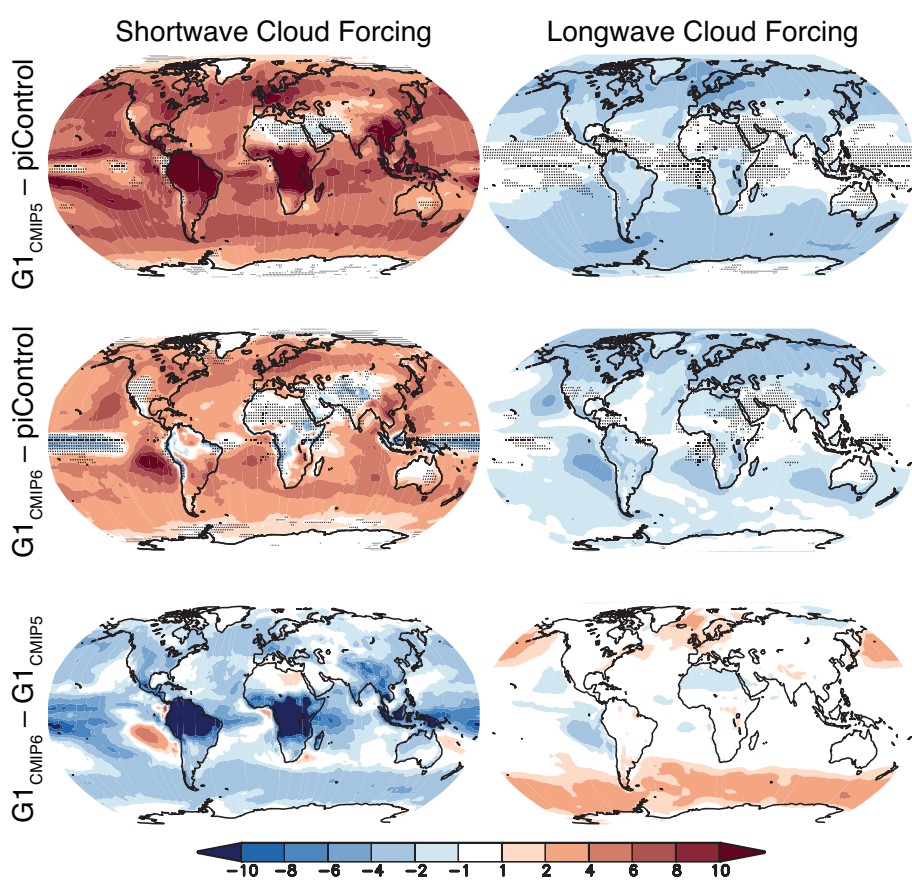

**Figure 5.** Surface shortwave (left) and longwave (right) cloud forcing (W m$^{-2}$) change from preindustrial for the CMIP5 (top) and CMIP6 (middle) ensembles, as well as the ensemble differences (bottom). Cloud forcing is measured as all-sky minus clear-sky radiative flux. All shaded values are ensemble means. Lack of stippling indicates agreement on the sign of the values across at least 75% of the models.

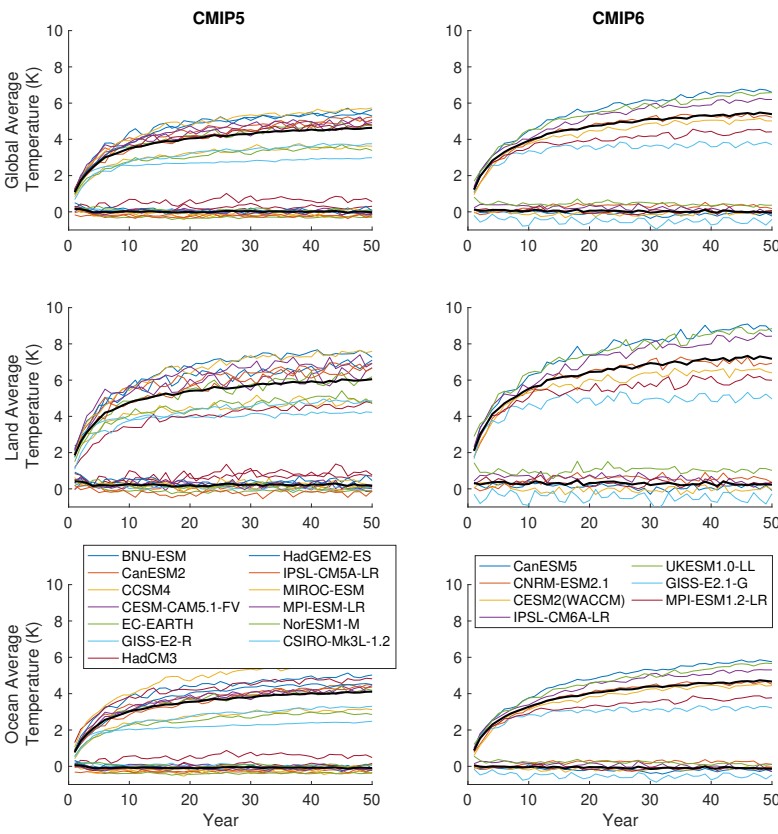

**Figure 6.** Global mean (top), land mean (middle), and ocean mean (bottom) temperature change (K) for the CMIP5 (left) and CMIP6 (ensembles). Thin colored lines are individual models, and thick black lines are model means. In all panels, the upper cluster of lines is the abrupt4xCO2 simulation, and the lower cluster of lines (approximately zero temperature change for the entire simulation) is experiment G1.



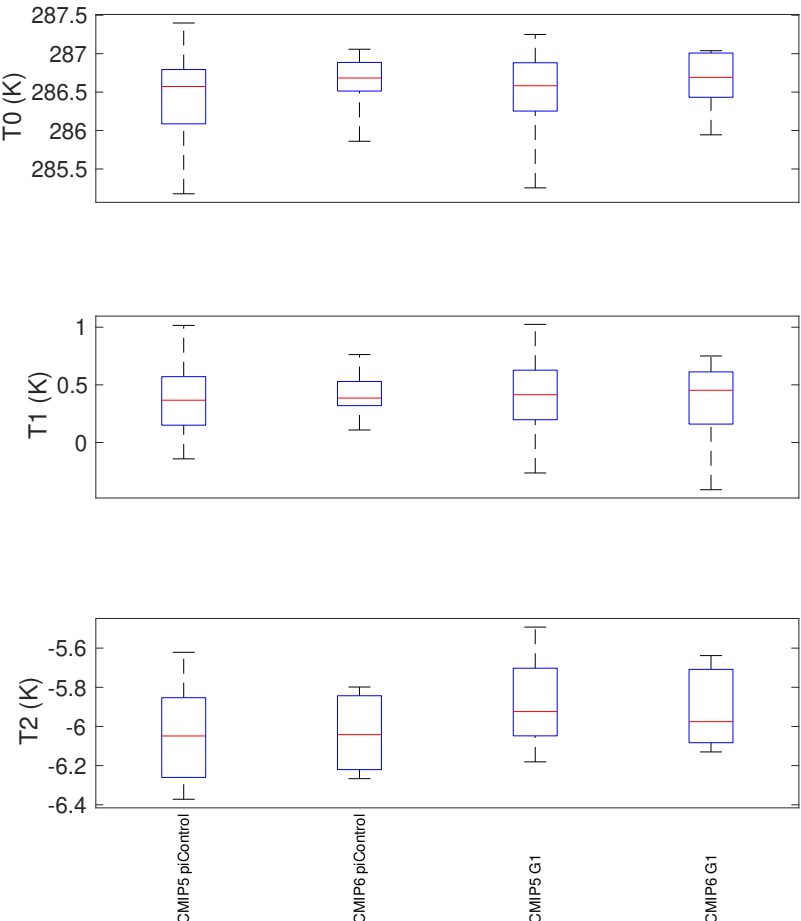

**Figure 7.** Ensemble ranges for global mean temperature ($T_0$), the interhemispheric temperature gradient ($T_1$), and the equator-to-pole temperature gradient ($T_2$), as defined in Equation 1 (Ban-Weiss and Caldeira, 2010; Kravitz et al., 2016). Red lines indicate ensemble medians, blue boxes are the inter-quartile ranges, and black whiskers indicate total ranges.





**Figure 8.** Ensemble average temperature changes (K) for G1 (as compared to the preindustrial control) for CMIP5 (top) and CMIP6 (middle), as well as the difference (G1$_{CMIP6}$ minus G1$_{CMIP5}$, bottom panel). In the top two panels, stippling indicates regions where fewer than 75% of the models in their respective ensembles agree on the sign of the response.

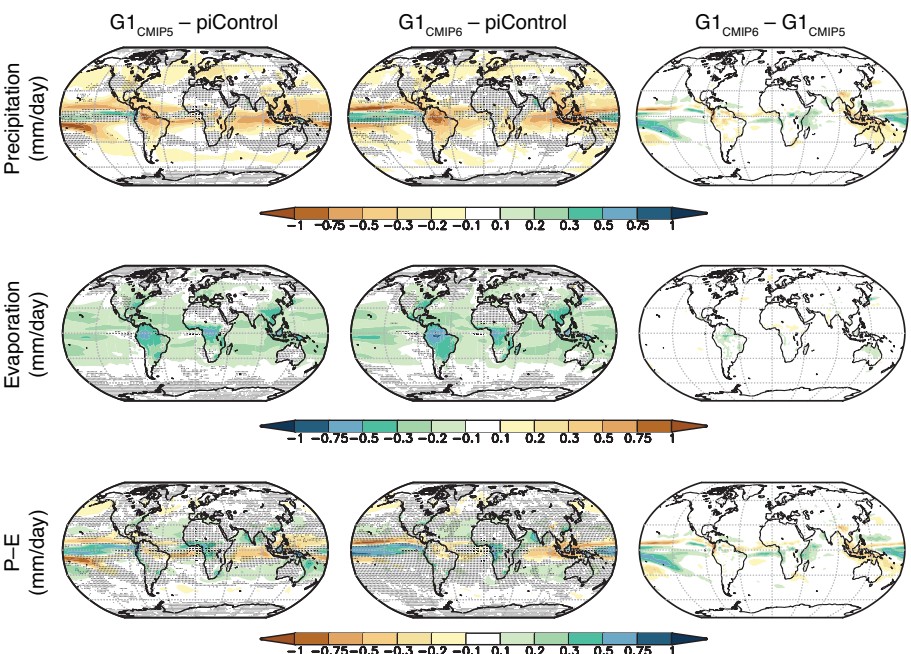

**Figure 9.** Precipitation (top), evaporation (middle), and precipitation minus evaporation (P–E; bottom) change from preindustrial for the CMIP5 (left) and CMIP6 (middle) ensembles, as well as the ensemble differences (right). All shaded values are ensemble means. Lack of stippling in the left and middle panels indicates agreement on the sign of the values across at least 75% of the models.

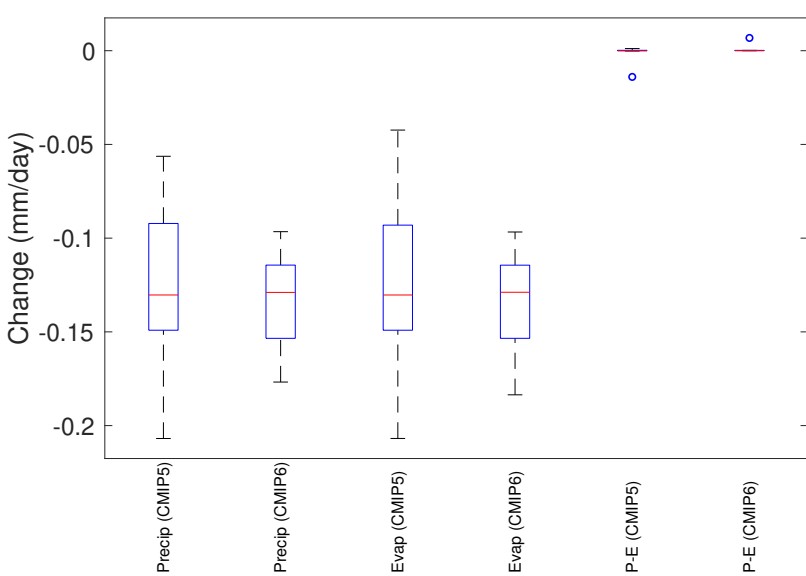

**Figure 10.** Global mean ensemble median (red lines), inter-quartile (blue boxes), and ranges (black whiskers or, for P–E one blue circle indicating an extreme outlier) for the hydrological quantities shown in Figure 9 for both the CMIP5 and CMIP6 ensembles.

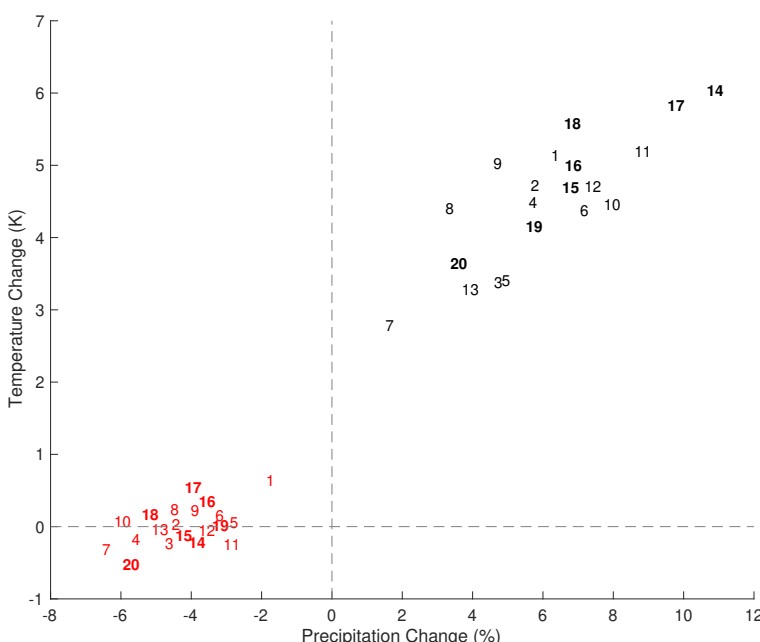

**Figure 11.** Average (years 11-50) temperature (y-axis; K) and precipitation (x-axis; %) change for each model in this study. Numbers indicate the model number (listed in Table 1, first column). Black numbers are for abrupt4xCO2, and red numbers are for G1. Bolded numbers are for CMIP6





**Figure 12.** Terrestrial net primary productivity (kg C m$^{-2}$ y$^{-1}$) for the CMIP5 (top) and CMIP6 (middle) ensembles, as well as the ensemble differences (bottom). All shaded values are ensemble means. Lack of stippling indicates agreement on the sign of the values across at least 75% of the models.