# Peer review of "Comparing different generations of idealized solar geoengineering simulations in the Geoengineering Model Intercomparison Project (GeoMIP)"

_Atmospheric Chemistry and Physics, 2020_

## Referee Comment (RC1) · Anonymous Referee #3 · 8 Oct 2020

This paper compares how two generations of Geoengineering models perform in a 50 year G1 experiment, where CO2 is instantaneously quadrupled and at the same time, insolation is reduced so the net TOA radiative flux is essentially unchanged. Key aspects of the CMIP5 vs. CMIP6 model ensemble results remain unchanged.

This is a worthwhile exercise for identifying where to focus more detailed analysis in future studies. However, it should also be stated clearly that consistency among ensembles of two generations of models demonstrates only that the differences do not significantly affect the results, at least when aggregated into ensemble averages; it by

no means shows the results to be robust in any deeper sense. (E.g., lines 186-188) This is especially important, as there are policymakers who cite the model results as support for proposed action in response to a possible severe climate crisis. The statement in lines 205-207 does not seem to be enough; this point also needs to be made much earlier in the paper in my opinion.

Without exploring more specifically the similarities and differences among the models, one cannot assess how significant the model diversity really is for generating the parameter values analyzed here. As such, this paper could go further in at least identifying which processes are most important for future focus (in addition to cloud parameterizations, which is already well known).

Notes

Figure 1. I can see that IPSL-CM6A-LR shows a long-term trend in temperature, as stated, but it seems CNRM has one too. UKESM1.0-LL has a jump at about 50 years that produces an overall pattern with the opposite sign. And if GISS shows trends in precipitation and evaporation, MPI seems to have these features as well.

Figure 1. For precipitation (and evaporation), two models are systematically below all the others, bringing the ensemble average to below the larger model cluster. This might require some explanation. For example, is there a fundamental unresolved issue in modeling precipitation, where a poorly constrained choice produces dramatically different results?

Lines 170-173. The fact that the CMIP6 models show less diversity does not mean that the uncertainty is lower. As more model inter-comparisons take place, it is not surprising that model behavior tends to converge. But to make a statement about model uncertainty requires critical tests, based on measurements at least of the processes involved.

Minor Notes

Table 2. The caption says Column 3 reports a fraction, whereas the column itself is labeled "%," which seems correct.

Figures 1 and 2. The colors assigned to the different models are not the same in these two figures, which seems unnecessarily confusing.

---

## Referee Comment (RC2) · Anonymous Referee #4 · 15 Dec 2020

This paper presents an interesting comparison between two generations of models performing a geoengineering experiment where quadrupling of $CO_2$ is offset by solar reduction. The goal of the study is to assess the consistency of model results between the two generations and the validity of the overall scientific conclusions.

This is attained applying standard statistical methods, in order to derive quantitative figures to support the conclusions. Although this study does not go into the details of specific models, but rather looks at the ensemble perspective, I think the topic is scientifically relevant and makes this study worthy of publication in ACP.

[Figure]

The paper is well written, concise and understandable. I have, however, a few minor remarks and suggestions for improvement that should be taken into account.

**Remarks**

L4: "This simulation is artificial", I think every model simulation is somewhat artificial, so maybe you could write "idealized".

L17: "Climate models remain the most promising tools...". Aren't they actually the only tool for that?

L34: What do you mean exactly with "agriculture"? Could you please be more specific?

L52: "unchanged from the baseline": does this mean that it has the same insolation as the 1850-1860 period of the piControl experiment? Please clarify.

L57-64: the trend in TOA net flux is not mentioned in this paragraph, although Fig. 1 shows it. Moreover, it would be interesting to have some numbers about the trends of the ensemble mean for each variable shown in Fig. 1.

Fig. 2: The legend hides some important parts of the plot, please consider shifting it to a different position.

Fig. 3: The brown line for the TOA net radiative flux change is probably the most interesting results in this figure but it's hardly visible. You may consider drawing it in the foreground above the other lines.

**Text corrections**

L35: I would replace "perfect" with "exact".

L38: I would replace "perfeclty" with "completely".

L43: I would replace "version of Earth System Models" with "version of the participating Earth System Models".

L58: I would replace "rare events or" with "rare events and".

L61: "by only analysing 50 years", do you mean "the first 50 years"?

L135: I think "or has warmer tropics" should be "nor has warmer tropics".

L199: It looks like punctuation is missing.

Fig. 1: Please add the mean to the legend (as in Fig. 2).

Fig. 3: I would replace "the difference" with "their difference".
* * *

---

## Author Comment (AC1) · 27 Dec 2020

Response to reviewer comments
Kravitz et al. (acp-2020-732)
Original review comments in normal text.  **Responses in bold.**

Anonymous Referee #3

This paper compares how two generations of Geoengineering models perform in a 50 year G1 experiment, where CO2 is instantaneously quadrupled and at the same time, insolation is reduced so the net TOA radiative flux is essentially unchanged. Key aspects of the CMIP5 vs. CMIP6 model ensemble results remain unchanged.
This is a worthwhile exercise for identifying where to focus more detailed analysis in future studies. However, it should also be stated clearly that consistency among ensembles of two generations of models demonstrates only that the differences do not significantly affect the results, at least when aggregated into ensemble averages; it by no means shows the results to be robust in any deeper sense. (E.g., lines 186-188) This is especially important, as there are policymakers who cite the model results as support for proposed action in response to a possible severe climate crisis. The statement in lines 205-207 does not seem to be enough; this point also needs to be made much earlier in the paper in my opinion.

**We thank the reviewer for their comments on our manuscript.  We agree with these points and have adjusted our language so as not to overstate our conclusions.  We have also made the point in lines 205-207 earlier in the manuscript.**

Without exploring more specifically the similarities and differences among the models, one cannot assess how significant the model diversity really is for generating the parameter values analyzed here. As such, this paper could go further in at least identifying which processes are most important for future focus (in addition to cloud parameterizations, which is already well known).

**We appreciate the reviewer's point.  We would like to acknowledge the difficulty of what the reviewer suggests.  Even though many models are related historically (e.g., Knutti et al., 2013), in some cases large parts of the model have been completely overhauled between versions, which does not provide a controlled enough experiment to identify which processes are most important for future focus.  Cloud parameterizations and land surface parameterizations were obvious ones (as the reviewer states), but beyond that we don't have enough information to even conduct informed speculation.  We found it interesting that despite these large differences, the two generations of models largely resulted in similar climate outcomes.  We agree that these points were not necessarily communicated as clearly as they could have been.  We have gone through the manuscript to address this.**

Notes

Figure 1. I can see that IPSL-CM6A-LR shows a long-term trend in temperature, as stated, but it seems CNRM has one too. UKESM1.0-LL has a jump at about 50 years that produces an overall pattern with the opposite sign. And if GISS shows trends in precipitation and evaporation, MPI seems to have these features as well.

**We have added a more thorough discussion of trends to the manuscript, including a new Table that provides quantitative estimates of trends.**

Figure 1. For precipitation (and evaporation), two models are systematically below all the others, bringing the ensemble average to below the larger model cluster. This might require some explanation. For example, is there a fundamental unresolved issue in modeling precipitation, where a poorly constrained choice produces dramatically different results?

**This is a fair point. There are no observations of geoengineering, so it's difficult to say exactly why these two models are different, although we can speculate a little bit. GISS-E2.1-G cools more than the other models (Figure 1) and thus would be expected to have greater reduction in hydrological cycle strength. At present we do not have an explanation for the IPSL-CM6A-LR results – although precipitation change is correlated with the global-scale atmospheric heating rate (approximated as the difference between net TOA and net surface radiative fluxes), there is no appreciable difference between the CMIP6 models.**

**In any case, we are reluctant to focus on these two because being an outlier doesn't necessarily mean being wrong. We have added some discussion of this to the manuscript.**

Lines 170-173. The fact that the CMIP6 models show less diversity does not mean that the uncertainty is lower. As more model inter-comparisons take place, it is not surprising that model behavior tends to converge. But to make a statement about model uncertainty requires critical tests, based on measurements at least of the processes involved.

**Our original language was unclear – this statement was about the findings of Arora et al. (2020). We have clarified this in the manuscript.**

Minor Notes

Table 2. The caption says Column 3 reports a fraction, whereas the column itself is labeled "%," which seems correct.

**That's a typo in the caption. Thanks for catching that.**

Figures 1 and 2. The colors assigned to the different models are not the same in these two figures, which seems unnecessarily confusing.

**Corrected. Thanks for catching that oversight.**

Anonymous Referee #4

This paper presents an interesting comparison between two generations of models performing a geoengineering experiment where quadrupling of CO2 is offset by solar reduction. The goal of the study is to assess the consistency of model results between the two generations and the validity of the overall scientific conclusions.
This is attained applying standard statistical methods, in order to derive quantitative figures to support the conclusions. Although this study does not go into the details of specific models, but rather looks at the ensemble perspective, I think the topic is scientifically relevant and makes this study worthy of publication in ACP.
The paper is well written, concise and understandable. I have, however, a few minor remarks and suggestions for improvement that should be taken into account.

**Thank you!**

Remarks
L4: "This simulation is artificial", I think every model simulation is somewhat artificial, so maybe you could write "idealized".

**Agreed.  Changed.**

L17: "Climate models remain the most promising tools...". Aren't they actually the only tool for that?

**Not entirely – one could gain information from natural analogues like volcanic eruptions, as well as physical understanding (with which the model helps).  Nevertheless, the reviewer makes a valid point, and we have updated the manuscript accordingly.**

L34: What do you mean exactly with "agriculture"? Could you please be more specific?

**This has been clarified.**

L52: "unchanged from the baseline": does this mean that it has the same insolation as the 1850-1860 period of the piControl experiment? Please clarify.

**We have clarified this in the manuscript.**

L57-64: the trend in TOA net flux is not mentioned in this paragraph, although Fig. 1 shows it. Moreover, it would be interesting to have some numbers about the trends of the ensemble mean for each variable shown in Fig. 1.

**Agreed.  We have added a discussion of this to the manuscript, including a new table.**

Fig. 2: The legend hides some important parts of the plot, please consider shifting it to a different position.

**Done.**

Fig. 3: The brown line for the TOA net radiative flux change is probably the most interesting results in this figure but it's hardly visible. You may consider drawing it in the foreground above the other lines.

**Done.**

Text corrections

L35: I would replace "perfect" with "exact".
L38: I would replace "perfeclty" with "completely".
L43: I would replace "version of Earth System Models" with "version of the participating Earth System Models".
L58: I would replace "rare events or" with "rare events and".
L61: "by only analysing 50 years", do you mean "the first 50 years"?
L135: I think "or has warmer tropics" should be "nor has warmer tropics".
L199: It looks like punctuation is missing.
Fig. 1: Please add the mean to the legend (as in Fig. 2).
Fig. 3: I would replace "the difference" with "their difference".

**All of these text corrections have been made.**

[revised manuscript text omitted]

The original G1 experiment was 50 years in length, whereas the CMIP6 version is 100 years in length to allow for better analyses of rare events  and to capture very slow responses. Comparison between the two ensembles necessitates only using the first 50 years, but we need to verify that this can be done without losing important longer-term evolution in features. Figures 1 and 2 look at G1 behavior over the entire 100-year period of the CMIP6 simulations to determine whether there is any drift or  steady state error that would not be revealed by only analyzing the first 50 years.  (Also see Table 2 for quantitative information.) Over years 11–100 of simulation, CNRM-ESM2.1 and IPSL-CM6A-LR  show greater than 0.1 K/decade in magnitude negative trends in temperature, and CESM2(WACCM) and UKESM1.0-LL show positive trends of similar magnitudes. This is despite no model showing a trend in net TOA radiative flux greater in magnitude than 0.02 W m$^{-2}$/decade. Beyond an initial transient period, CESM2(WACCM), CNRM-ESM2.1, and IPSL-CM6A-LR show approximately 0.06%/decade trends in precipitation and evaporation of the same sign as the temperature trends. Nevertheless, the differences in temperature and hydrological cycle change due to experiment G1 are orders of magnitude greater than the calculated values in Table 2. As such, we conclude that our choice to focus on the first 50 years of simulation does not appreciably affect our results.

Figure 2  shows that many of the models have low frequency variability that appears in the different regions plotted here. For the egion north of 30°N, IPSL-CM6A-LR  has a steadily increasing temperature value north of 30°N, possibly related to a slight trend in sea ice coverage (Boucher et al., 2020).

IPSL-CM6A-LR is also known to have a bicentennial oscillation, which could affect G1–piControl differences, depending on the baseline period used for subtraction. To verify that this oscillation is not impacting our results, we divided that model's 1200-year piControl run into 50-year chunks and computed the surface air temperature average for each of those chunks. The largest temperature found was 286.0339 K, and the smallest was 285.6384 K. The average over the entire ensemble was 285.8604 K. As such, using the mean of the entire ensemble versus matching the appropriate period in the bicentennial oscillation would have an impact on G1–piControl temperature by at most 0.22 K. Only averaging the first 100 years of the piControl run (which may be the best match to the period covered by G1) yields a temperature of 285.9084 K, which is 0.048 K different from the mean of the entire piControl run. As such, we conclude that this bicentennial oscillation is unlikely to have substantially influenced our findings.

Per the results in Figure 1, IPSL-CM6A-LR and GISS-E2.1-G appear to have a different responsiveness of the hydrological cycle to the combined $CO_2$-solar forcing than the other models. We are reluctant to attribute this feature to any potential shortcomings or lack of fidelity to observations because there are no observations of this type of experiment. Although these models are outliers, there is no evidential basis on which to assume they are more or less valid than the other models for this study.

Because the main focus of this paper is a comparison between the CMIP5 and CMIP6 generations of model results, we have opted for the following to aid comparisons:

- Since we are not evaluating any features that require 100 years of statistics, and the results do not show any appreciable time evolution of behavior after the first couple of years (see discussion above), we only evaluate the first 50 years of all simulations. All maps show changes over years 11-50, removing the initial transient period.

- We do not compare previous versions of individual models with current ones, instead only examining ensembles. Even though models may share similar development histories (e.g., atmosphere and ocean dynamical cores, convective parameterizations, radiative transfer modules, terrestrial biosphere and cryosphere; Knutti et al., 2013; Zelinka et al., 2020), there have been numerous developments in models in these areas (and others) between CMIP5 and CMIP6 such that in most cases a direct comparison would not be meaningful.

- We focus extensively on the G1 results and, with few exceptions, do not focus on the corresponding abrupt4xCO2 simulations. It has been well documented that the CMIP6 models tend to have higher climate sensitivities than the CMIP5 models (Flynn and Mauritsen, 2020; Meehl et al., 2020; Zelinka et al., 2020), so we do not wish to make conclusions that might be based on a form of selection bias.

- All lack of stippling on map plots, as in previous GeoMIP studies (e.g., Kravitz et al., 2013), indicates agreement on the sign of the response in at least 75% of models. Because $G1_{CMIP5}$ has more participating models than $G1_{CMIP6}$, this threshold provides some consistency across analyses of the ensembles. When plotting differences between the ensembles ($G1_{CMIP6}$–$G1_{CMIP5}$), there is no stippling, as it is difficult to meaningfully represent such differences between ranges. Aggregate differences between the two ensembles, as calculated using Welch's $t$-test or differences in stippled area, are discussed in Table 3.

For CMIP6, we analyzed one ensemble member for all experiments except for CanESM5 (G1), CNRM-ESM2.1 (abrupt4xCO2 and G1), and UKESM1.0-LL (G!).

[revised manuscript text omitted]
 (and indeed the present setup does not allow for a controlled experiment to rigorously test structural uncertainty), we show that the ensemble spread of total terrestrial NPP is smaller in CMIP6 than in CMIP5. This result is consistent with the recent assessment of carbon cycle feedbacks conducted by Arora et al. (2020),  who showed that the CMIP6 ensemble has reduced overall  model spread in the land carbon cycle to rising $CO_2$ compared to their CMIP5 predecessors.

**4 Discussion and Conclusions**

Based on the results presented here, model response to G1 has not changed substantially between CMIP5 and CMIP6, despite numerous changes to models between the two generations, including an increase in climate sensitivity. The sign of residual climate impacts (for example in temperature) are in better agreement in CMIP5 than CMIP6 (Table 3 shows a difference in stippled area between the two ensembles), but this could be a function of the smaller ensemble size in CMIP6. Energetics, temperature, and the hydrological cycle are qualitatively and quantitatively similar in both ensemble means and ensemble ranges, although these variables are somewhat related, so we might expect them to all portray a similar picture. Notable

differences do exist in shortwave cloud forcing and NPP, particularly in Amazonia, Africa, and Australia, which are also regions of inter-ensemble difference in precipitation.

From these findings, we can conclude that results obtained over the past 20 years of study have not been overturned by the latest round of simulations. All of the major ensemble differences highlighted above deal with clouds and land surface modeling, both of which are difficult to model and are necessarily highly parameterized. The conclusions that are based on more fundamental knowledge, such as column energetics (in the case of the hydrological cycle), are relatively robust to structural uncertainty, in so far as this study adequately captures representative variations in structural uncertainty. This lends confidence to our conclusions, especially regarding robustness to uncertainty, about the broad climate effects from solar geoengineering methods that can be accurately represented via solar dimming.

We also conclude that the models used in CMIP5 are not obviously biased or inferior as compared to CMIP6. While improvements have been made in the CMIP6 generation of models, and those models are likely better for representing numerous features of the present-day climate that may be important for studies of geoengineering, there are many aspects of climate that are well represented by earlier models. In some cases, more robust analyses may be enabled by augmenting ensemble sizes with archived output from earlier generations of CMIP models.

Many of the broad features of solar geoengineering with sulfate aerosols can be represented by a reduction in solar constant (e.g., Niemeier et al., 2013; Kalidindi et al., 2015). However, the more subtle changes that derive from complex response to stratospheric aerosol heating  (for example, consequences of stratospheric heating like the positive wintertime North Atlantic Oscillation; Simpson et al., 2019; Jones et require detailed assessments with state-of-the-art aerosol microphysical schemes. This is particularly important for understanding regional and seasonal solar geoengineering (Kravitz et al., 2017; Visioni et al., 2019). Such detailed microphysical calculations can only be simulated in a small number of models; in the case of Jones et al. (2021), only two models were available. While simple G1-style experiments enable a robust multi-model ensemble analysis, they cannot capture details that depend on microphysics. We emphasize the importance of a variety of modeling approaches to understand solar geoengineering, particularly the role of model uncertainty in conclusions about solar geoengineering.

There are numerous aspects of physical climate that we did not evaluate, nor did we pursue analyses beyond physical climate, including many other aspects of natural science, social science, the humanities, governance, justice, or ethics (to name a few important areas). Moreover, we emphasize that experiment G1 is an idealized experiment aimed at understanding physical climate response to combinations of large forcings and should not be interpreted as a realistic or policy-relevant scenario of geoengineering. A holistic assessment of the consequences of geoengineering, particularly of more policy-relevant scenarios, would certainly need to take these numerous aspects into account. Nevertheless, based on the results presented here, results for geoengineering across several important metrics appear to be robust to some amount of structural uncertainty. This lends confidence to some conclusions drawn from global climate models regarding solar geoengineering.

*Data availability.* All CMIP5 and CMIP6 output, including the respective GeoMIP simulations, is available via the Earth System Grid Federation (https://esgf-node.llnl.gov/projects/esgf-llnl/) or by contacting the respective modeling groups responsible for the output. For
235 CMIP6 output, seeTable 1 data citations.

*Author contributions.* BK, OB, JNSC, JH, AJ, TL, PN, UN, RS, and ST contributed model output. BK performed the analysis. BK, DGM, and DV wrote the manuscript with all coauthors.

*Competing interests.* None.

*Acknowledgements.* We acknowledge the World Climate Research Programme, which, through its Working Group on Coupled Modelling,
240 coordinated and promoted CMIP. We thank the climate modeling groups for producing and making available their model output, the Earth System Grid Federation (ESGF) for archiving the data and providing access, and the multiple funding agencies who support CMIP6 and ESGF. We also thank all participants of the Geoengineering Model Intercomparison Project and their model development teams. Support for B.K. was provided in part by the National Science Foundation (NSF) through agreement CBET-1931641, the Indiana University Environmental Resilience Institute, and the Prepared for Environmental Change Grand Challenge initiative. The Pacific Northwest National Laboratory
245 is operated for the US Department of Energy by Battelle Memorial Institute under contract DE-AC05-76RL01830. Resources supporting this work were provided by the NASA High-End Computing (HEC) Program through the NASA Center for Climate Simulation (NCCS) at Goddard Space Flight Center. A.R. is supported by NSF grants AGS-1617844 and AGS-2017113. J.H. and A.J. were supported by the Met Office Hadley Centre Climate Programme funded by BEIS and Defra. 
[revised manuscript text omitted]

Jones, A., Haywood, J., Jones, A., Tilmes, S., Robock, A., and Kravitz, B.: North Atlantic Oscillation response in GeoMIP experiments G6solar and G6sulfur: why detailed modelling is needed for understanding regional implications of solar radiation management, Atmos. Chem. Phys., accepted, 2021.

[revised manuscript text omitted]

**Table 3.** Ensemble differences between the CMIP5 and CMIP6 ensembles for each variable evaluated in this study (left column). Column 2 indicates the difference between the ensembles in how much of the Earth's surface is not stippled (more than 75% of models agree on the sign of the response; negative values indicate that CMIP6 has less unstippled area than CMIP5). Column 3 indicates the percent of the Earth's surface for which the CMIP5 ensemble is statistically different from the CMIP6 ensemble, based on 95th percentile confidence intervals from Welch's $t$-test.

[revised manuscript text omitted]

---

## Referee Report (RR1)

January 2021

Review 2 of Kravitz et al., ***Comparing different generations of idealized solar geoengineering simulations in the Geoengineering Model Intercomparison Project (GeoMIP)***

**Overall Notes**

This is my second review of the paper, which compares how two generations of Geoengineering models perform in a 50-year G1 experiment, where CO2 is instantaneously quadrupled and at the same time, insolation is reduced so the net TOA radiative flux is essentially unchanged. Key aspects of the CMIP5 vs. CMIP6 model ensemble results remain unchanged.

The authors have addressed my questions on the Results section in the previous version of the paper and have mitigated some overstatements of the conclusions. However, there remain a few key places in the Abstract, Introduction, and Conclusions sections where I think the way the results are framed still crosses the line. For example, in the Abstract, it states: "*We conclude that despite numerous structural differences and uncertainties in models … broad conclusions about the climate response to global solar dimming remain robust.*" Yet the previous sentence says: "*… the only major differences involve highly parameterized and uncertain processes, such as cloud forcing or terrestrial net primary productivity.*" Cloud forcing represents a major uncertainty in nearly all aspects of climate prediction and is possibly the leading uncertainty overall. The scope of the current paper does not include a detailed examination of the way cloud forcing is parameterized in different models – whether they all use similar parameterizations, or whether they capture the full range of plausible parameterizations, or something else.

Fair enough. But as such, the paper does not show that the actual uncertainty in cloud forcing has minimal effect on the model results. One can say that available GeoMIP model simulations of the climate response to global solar dimming are in aggregate mostly consistent between iterations 5 and 6. However, one cannot claim that the actual climate response to global solar dimming is represented robustly based on the model comparisons provided. This distinction is subtle but important, especially as geoengineering is also a topic for policy makers, who might not appreciate the limitations of the models, and as a consequence, might make decisions based on assuming that the model results robustly reflect nature. (This note also applies to lines 202-204 and lines 225-227 in the Conclusions. I appreciate that there are now also some more evenhanded statements in the conclusions, but the texts in these two places remain as overstatements of the type discussed above.)

**Additional Notes**

Lines 16-17. "*… replicating the mechanisms that cause cooling after large volcanic eruptions (Robock, 2000).*" There are also some important differences between the proposed, continual geoengineering stratospheric injection and natural volcanic injections. This point is also made in some of Alan's papers.

Lines 27-28. "*Simulations of solar geoengineering with solar reduction have long shown that solar geoengineering would cool the planet, offsetting global warming.*" However, the "offsetting" occurs only in a global-average sense. The following few sentences leave the impression that solar geoengineering would reverse changes in the hydrologic cycle, the cryosphere, extreme events, vegetation, circulation, etc., with the offsets just not necessarily being "exact" on a regional basis, not being able to "completely" offset climate change from greenhouse gases. This again understates the uncertainties – there might be far more "losers" than "winners," for example, and the models are by no means good enough to draw strong conclusions about this. The current paper could help, if this point is framed differently.

Lines 191-193. "*The sign of residual climate impacts (for example in temperature) are in better agreement in CMIP5 than CMIP6 (Table 3 shows a difference in stippled area between the two ensembles), but this could be a function of the smaller ensemble size in CMIP6.*" It would be difficult to actually demonstrate this conjecture statistically, given the small numbers of models involved, and the current paper does not attempt it. An alternative interpretation of the observation would be that the factors affecting the signs of residual climate impacts are not well enough understood for the CMIP6 models to show improvement over CMIP5. Again, the current paper could help…

---

## Author Response (AR2)

Response to reviews
Kravitz et al.
Original reviewer comments in normal text.  **Responses in bold.**
* * *
Review 2 of Kravitz et al., *Comparing different generations of idealized solar geoengineering simulations in the Geoengineering Model Intercomparison Project (GeoMIP)*

Overall Notes

This is my second review of the paper, which compares how two generations of Geoengineering models perform in a 50-year G1 experiment, where $CO_2$ is instantaneously quadrupled and at the same time, insolation is reduced so the net TOA radiative flux is essentially unchanged. Key aspects of the CMIP5 vs. CMIP6 model ensemble results remain unchanged.

The authors have addressed my questions on the Results section in the previous version of the paper and have mitigated some overstatements of the conclusions. However, there remain a few key places in the Abstract, Introduction, and Conclusions sections where I think the way the results are framed still crosses the line. For example, in the Abstract, it states: "*We conclude that despite numerous structural differences and uncertainties in models ... broad conclusions about the climate response to global solar dimming remain robust.*" Yet the previous sentence says: "*... the only major differences involve highly parameterized and uncertain processes, such as cloud forcing or terrestrial net primary productivity.*" Cloud forcing represents a major uncertainty in nearly all aspects of climate prediction and is possibly the leading uncertainty overall. The scope of the current paper does not include a detailed examination of the way cloud forcing is parameterized in different models – whether they all use similar parameterizations, or whether they capture the full range of plausible parameterizations, or something else.

**We thank Ralph Kahn for the insightful comments and for multiple rounds of revisions.**

Fair enough. But as such, the paper does not show that the actual uncertainty in cloud forcing has minimal effect on the model results. One can say that available GeoMIP model simulations of the climate response to global solar dimming are in aggregate mostly consistent between iterations 5 and 6. However, one cannot claim that the actual climate response to global solar dimming is represented robustly based on the model comparisons provided. This distinction is subtle but important, especially as geoengineering is also a topic for policy makers, who might not appreciate the limitations of the models, and as a consequence, might make decisions based on assuming that the model results robustly reflect nature. (This note also applies to lines 202-204 and lines 225-227 in the Conclusions. I appreciate that there are now also some more evenhanded statements in the conclusions, but the texts in these two places remain as overstatements of the type discussed above.)

**This is an important point, and we acknowledge that we did not communicate (un)certainty as well as we should have.  We have made the following changes:**

- **In the abstract, instead of speaking about the past 20 years of simulation, we have changed to only describe the past two generations of models.**
- **In the abstract, we have removed discussions of robustness and have adopted the reviewer's phrasing that the models are consistent in their aggregate climate response.**
- **In lines 202-204, we have similarly removed mentions of robustness and have changed the description to more accurately represent what was simulated and analyzed.**
- **In lines 225-227, we have removed mentions of robustness and instead describe the results as consistent across different generations of models.**

Additional Notes

Lines 16-17. *"... replicating the mechanisms that cause cooling after large volcanic eruptions (Robock, 2000)."* There are also some important differences between the proposed, continual geoengineering stratospheric injection and natural volcanic injections. This point is also made in some of Alan's papers.

**Good point.  We have modified the language in the text and added a citation to Robock et al. (2013).**

Lines 27-28. "*Simulations of solar geoengineering with solar reduction have long shown that solar geoengineering would cool the planet, offsetting global warming*." However, the "offsetting" occurs only in a global-average sense. The following few sentences leave the impression that solar geoengineering would reverse changes in the hydrologic cycle, the cryosphere, extreme events, vegetation, circulation, etc., with the offsets just not necessarily being "exact" on a regional basis, not being able to "completely" offset climate change from greenhouse gases. This again understates the uncertainties – there might be far more "losers" than "winners," for example, and the models are by no means good enough to draw strong conclusions about this. The current paper could help, if this point is framed differently.

**Good point – this statement only applies to the global average.  We have qualified the language to better reflect the model results.**

Lines 191-193. "*The sign of residual climate impacts (for example in temperature) are in better agreement in CMIP5 than CMIP6 (Table 3 shows a difference in stippled area between the two ensembles), but this could be a function of the smaller ensemble size in CMIP6.*" It would be difficult to actually demonstrate this conjecture statistically, given the small numbers of models involved, and the current paper does not attempt it. An alternative interpretation of the observation would be that the factors affecting the signs of residual climate impacts are not well enough understood for the CMIP6 models to show improvement over CMIP5. Again, the current paper could help...

**Good point.  We have included this alternate interpretation in the manuscript.**